# CopRA: A Progressive LoRA Training Strategy

**Zhan Zhuang**[1,2]    **Xiequn Wang**[1]    **Yulong Zhang**[3]    **Wei Li**[1]
**Yu Zhang**[1,*]    **Ying Wei**[3,*]

[1]Southern University of Science and Technology
[2]City University of Hong Kong    [3]Zhejiang University
12250063@mail.sustech.edu.cn
{zhangylcse, ying.wei}@zju.edu.cn
{wangxiequn, li.wei.ml.619, yu.zhang.ust}@gmail.com

## Abstract

Low-Rank Adaptation (LoRA) is a parameter-efficient technique for rapidly fine-tuning foundation models. In standard LoRA training dynamics, models tend to converge to a local optimum near the initialization quickly. However, this local optimum may not be ideal for out-of-distribution data or tasks such as merging and pruning. In this work, we introduce a novel progressive training strategy for LoRA that incorporates random layer dropping without incurring additional training costs. This strategy also optimizes the Shapley value of LoRA parameters in each layer, treating each layer as a player in a cooperative game. We refer to this method as **Coop**erative Lo**RA** (**CopRA**). Our experimental results demonstrate that parameters trained with CopRA exhibit linear mode connectivity, which enables efficient model merging. This also paves the way for federated learning and multi-task learning via LoRA merging. Additionally, by optimizing the Shapley value, CopRA shows superior performance in pruning tasks.

*"The strength of the team is each individual member. The strength of each member is the team."*

– Phil Jackson

## 1 Introduction

Understanding and interpreting neural network landscapes is critical and challenging in deep learning. Since neural networks are over-parameterized and highly non-convex, there exists multiple near-optima [18, 5]. Despite all the local optima with similar performance [5], the inherent training dynamics of gradient-based optimization algorithms cause each parameter to remain close to its initialization point [7, 3]. Recent investigations into mode connectivity [10, 6] and neuron permutation symmetries [2, 15, 28] have revealed that different local minima are not isolated and can be rebasin.

Advancements in foundation models have been significantly enhanced by low-rank adaptation (LoRA) [11], which modifies the weights of each existing layer using low-rank matrices. To improve model generalization and fusion, the composability of LoRA modules has been investigated [12]. Linear fusion [12, 25], illustrated in Fig. 1(a), is a straightforward method that preserves the low-rank characteristics of updated weights and does not require additional storage during inference. Linear mode connectivity (LMC) [9, 30], which reveals that different local minima can be connected linearly in the loss landscape, is a crucial property for linear fusion. However, previous works [27, 23] have shown that standard LoRA training may not achieve LMC, leading to poor accuracy in merging.

---

[*]Corresponding authors.

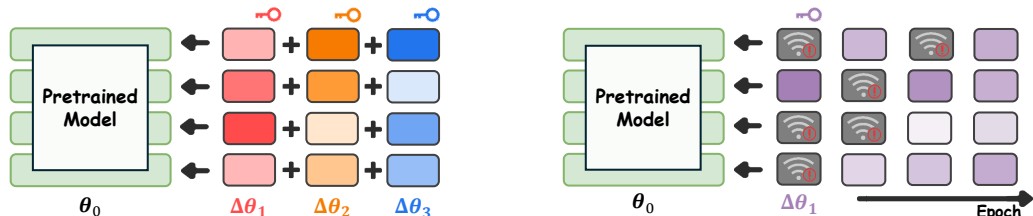

Figure 1: Illustration of (left) **LoRA merging** and (right) **Training strategy of CopRA**. Different colors represent parameters from different seeds or tasks, while grey indicates inactive parameters.

In this paper, we propose a novel training strategy for LoRA that facilitates the exploration of local optima exhibiting LMC. Building on the findings from the layer-wise linear mode connectivity [1], we observe that the averaging barrier of individual layers is minimal, suggesting that linear interpolation between two local optima remains optimal when training a single LoRA layer. Our strategy randomly determines the participation of each LoRA module during training based on an increasing probability, creating uncertainty in the subset of selected modules.

In the early training stages, fewer modules are optimized, resulting in a limited number of optimal solutions that are more likely to satisfy (layer-wise or group-wise) LMC. As training progresses, the number of updated LoRA modules per optimization iteration increases, enabling a broader exploration toward the global optimum and improved accuracy. Ultimately, this lazy training ensures that the final solution remains close to the preceding local optima, preserving the LMC property.

## 2 Methodology

**Notation.**  In this paper, for simplicity and generality, we represent the foundation model as an $L$-layer fully connected neural network, even though it is actually based on the Transformer architecture [22]. We denote the model as $f(\cdot; (\boldsymbol{W}_l)_{l=1}^L)$, where $(\boldsymbol{W}_l)_{l=1}^L$ represents the set of weight matrices for all layers. To keep the notation simple, we omit biases and activation functions, as LoRA only modifies the weight matrices. Given an input $\boldsymbol{x}_0$, the model's output $\hat{\boldsymbol{y}}$ can be expressed as:

$$\hat{\boldsymbol{y}} = f(\boldsymbol{x}_0; (\boldsymbol{W}_l)_{l=1}^L) = f_L(\boldsymbol{W}_L, \boldsymbol{x}_{L-1}), \quad \text{where} \quad \boldsymbol{x}_l = f_l(\boldsymbol{W}_l, \boldsymbol{x}_{l-1}), \ \forall l \in 1, \ldots, L-1. \quad (1)$$

Here, $\boldsymbol{x}_l$ serves as both the input to layer $l$ and the intermediate feature at that layer. LoRA employs two low-rank matrices to compute updates to the weight matrix: $\boldsymbol{A} \in \mathbb{R}^{r \times n}$ and $\boldsymbol{B} \in \mathbb{R}^{m \times r}$, where $r \ll \min(m, n)$. The weight matrix update is computed as $\Delta \boldsymbol{W} = \alpha \boldsymbol{B} \boldsymbol{A} \in \mathbb{R}^{m \times n}$, where $\alpha$ is a scaling factor. Thus, the fine-tuned model can be represented as $f(\cdot; (\boldsymbol{W}_l + \Delta \boldsymbol{W}_l)_{l=1}^L)$. To evaluate the model's performance, we use the cross-entropy (CE) loss, denoted as $\ell(\hat{\boldsymbol{y}}, \boldsymbol{y})$.

### 2.1 Training Strategy

In this section, we formally introduce our method. As illustrated in Fig. 1(b), we propose a progressive training strategy with random adapter dropping. The objective is formulated as follows:

$$\min_{(\Delta \boldsymbol{W}_l)_{l=1}^L} \ell(\hat{\boldsymbol{y}}, \boldsymbol{y}), \quad \text{where} \quad \hat{\boldsymbol{y}} = f(\boldsymbol{x}_0; (\boldsymbol{W}_l + \delta_l \Delta \boldsymbol{W}_l)_{l=1}^L) \text{ and } \delta_l \sim \text{Bernoulli}(p) \quad (2)$$

In this formulation, $(\delta_l)_{l=1}^L$ are independent random variables sampled from a Bernoulli distribution with parameter $p$, which increases progressively during training. Specifically, for a total of $T$ steps, at any given step $t$, the probability $p$ is defined as $\min\{\frac{4t}{3T}, 1\}$. The training process consists two stages:

- In the initial three-quarters of training ($t < \frac{3T}{4}$), $p$ gradually increases from 0 to 1.
- In the final quarter ($t \geq \frac{3T}{4}$), $p$ is set to 1, ensuring all LoRA layers are activated. At this stage, the training objective becomes equivalent to standard LoRA training.

This progressive changing objective approximates a multi-level optimization process, initially optimizing with a higher drop rate and then transitioning to a lower rate. As the number of trainable parameters increases, the optimal LoRA parameters for each layer expand within the weight space. With decreasing learning rates and the inertia of neural network parameters, this strategy encourages the exploration of new local optima near those identified in earlier training stages. Therefore, the final model will preserve the properties established initially, which enables linear mode connectivity.

## 2.2 A Cooperative Game Perspective

To gain deeper insights into this training strategy, we can view each LoRA layer as a player in a cooperative game, enabling structured analysis of individual contributions to overall model performance.

In general, a cooperative game can be represented by the pair $(N, v)$, where: $N = \{1, ..., n\}$ denotes the set of players (in this case, the individual LoRA layers); $v : 2^N \to R$ is the characteristic function. For any subset $C$, $v(C)$ quantifies the collective "reward" that the players in $C$ can achieve by working together. The Shapley value [19] offers a method to fairly allocate the total gain among all players based on their contributions. The proposed method helps evaluate each LoRA layer's marginal contribution by considering all possible layer combinations. Additionally, we show that our method approximates the optimization of the Shapley value for each LoRA layer.

Mathematically, directly computing the Shapley value is computationally expensive. To approximate it efficiently, one approach is the multilinear extension [16], where the Shapley value is calculated as:

$$\varphi_i(v) = \int_0^1 e_i(q)dq, \quad e_i(q) = \mathbb{E}\left[v\left(E_i \cup i\right) - v\left(E_i\right)\right]. \tag{3}$$

In our method, $E_i$ denotes a random subset of LoRAs excluding $i$, where each LoRA is selected with probability $q$. The term $e_i(q)$ captures the expected marginal contribution of LoRA $i$. This expectation is estimated through sampling, without considering changes in the learning rate, aligning with our optimization objective in Eq. 2. After training, we estimate the Shapley values for each layer, and the experimental results presented in Fig. 2 further validate our inference.

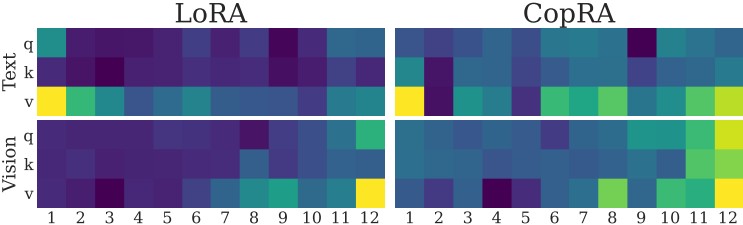

Figure 2: Approximated Shapley value of each layer.

## 3 Results

We adopt the setting of CLIP-LoRA [24] with 11 datasets [29] as the benchmark for merging evaluation. To further demonstrate the advantage of our method, we select *dtd* [4] and *ucf101* [20] for other evaluation. We use the default configuration with a rank $r = 2$ and ViT-B/16 as the backbone. The learning rate is set to 5e-4, and each task is run with five random seeds.

### 3.1 CopRA Excels in Merging due to Linear Mode Connectivity

There are two merging strategies [27, 23], fusion [12] and mixture. The updates can be formulated as:

$$\Delta \boldsymbol{W}_f = (\alpha \boldsymbol{B}_1 + (1 - \alpha)\boldsymbol{B}_2)(\alpha \boldsymbol{A}_1 + (1 - \alpha)\boldsymbol{A}_2) \quad \text{(Fusion)}$$
$$\Delta \boldsymbol{W}_m = \alpha \Delta \boldsymbol{W}_1 + (1 - \alpha)\Delta \boldsymbol{W}_2 = \alpha \boldsymbol{B}_1 \boldsymbol{A}_1 + (1 - \alpha)\boldsymbol{B}_2 \boldsymbol{A}_2 \quad \text{(Mixture)} \tag{4}$$

We adopt the fusion strategy for evaluation because it preserves the low-rank structure, even though the accuracy of the mixture (stacking) has been shown to be higher [27, 23]. Due to the equivariance within the adapter, we introduce a learnable invertible matrix $\boldsymbol{P}$ via SVD decomposition and leading to the formulation $\Delta \boldsymbol{W}_2 = (\boldsymbol{B}_2 \boldsymbol{P})(\boldsymbol{P}^{-1} \boldsymbol{A}_2)$. We then propose **LoRA align (LA)**, which minimizes an upper bound $\Delta_{\text{upper}}$ of the difference between the two merging methods, defined as:

$$\|\Delta \boldsymbol{W}_f - \Delta \boldsymbol{W}_m\|_2 \le \alpha(1 - \alpha)\left(\|\boldsymbol{B}_1 - \boldsymbol{B}_2 \boldsymbol{P}\|_2 + \|\boldsymbol{A}_1 - \boldsymbol{P}^{-1} \boldsymbol{A}_2\|_2\right) = \Delta_{\text{upper}} \tag{5}$$

Results presented in Fig. 3 illustrate the accuracy landscape with various interpolation coefficients, demonstrating the linear mode connectivity of CopRA. When the interpolation ratio is 0.5, meaning that the two models are evenly merged, we observe that CopRA achieves a significant improvement compared to LoRA. With LA, accuracy is further improved, suggesting that CopRA operates not within the adapter but across layers. However, due to the additional cost of LA's backpropagation, we do not employ LA in the following section.

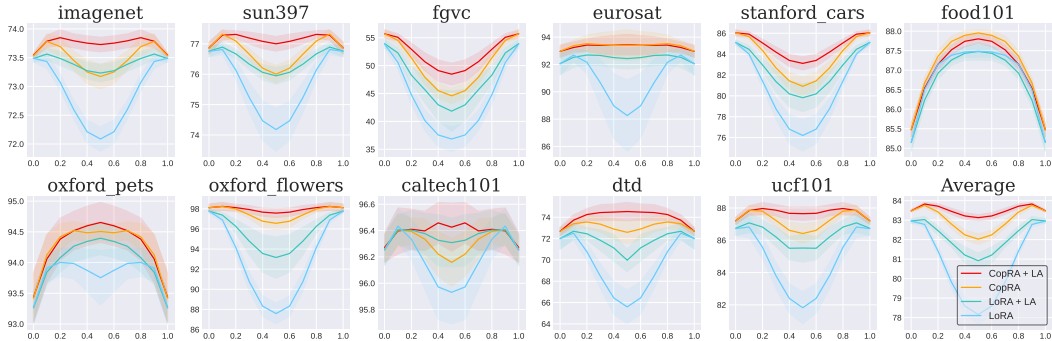

Figure 3: **Visualization of accuracy landscape across different methods for the CLIP datasets.** The X-axis represents the interpolation coefficient, while the Y-axis indicates accuracy (%).

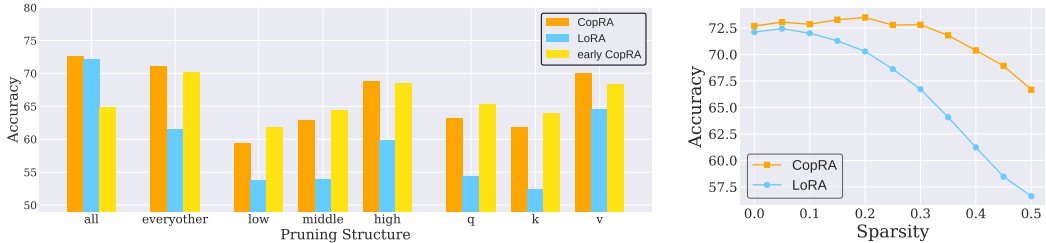

Figure 4: (left) **Structured pruning to various components**, including layers (every other, low, middle, high) and attention elements. (right) **Unstructured pruning with varying levels of sparsity.**

**Federated Learning and Multi-Task Learning.** Two important applications of CopRA are the aggregation of LoRAs in federated learning (FL) [14, 25, 23] and multi-task learning (MTL) [26, 27]. In FL, the dataset is randomly split into five subsets, each assigned to an independent client that trains its sub-model using different seeds. The server then merges the five trained sub-models. As shown in Tab. 1 (left), the accuracy of clients using LoRA and CopRA on two datasets is similar, while CopRA achieves superior aggregation.

Table 1: Accuracies (%) on the *dtd* and *ucf101* datasets under FL and MTL settings.

| | | FL | | MTL | |
| | | *dtd* | *ucf101* | *dtd* | *ucf101* |
|---|---|---|---|---|---|
| origin | LoRA | 69.15 | 85.18 | 72.22 | 86.06 |
| | CopRA | 69.16 | 85.61 | 72.80 | 87.39 |
| merge | LoRA | 54.37 | 73.01 | 58.58 | 78.77 |
| | CopRA | **64.07** | **79.25** | **64.88** | **81.42** |

In MTL, we fuse models trained on two different datasets and evaluate the merged model. The results in Tab. 1 (right) indicate that CopRA outperforms standard LoRA in terms of merging performance.

### 3.2 CopRA Provides a Simple Way to Prune

In this section, we highlight the advantages of CopRA in pruning tasks. Although LoRA already uses a relatively small number of parameters, further reducing redundancy is still beneficial. CopRA involves randomly skipping LoRA layers during training, similar to the random structured dropout [8] and stochastic depth [13]. This technique has shown promising results in sub-selection pruning strategies, making CopRA particularly well-suited for pruning.

Fig. 4 presents the results for both structured pruning and unstructured pruning of LoRA and CopRA. In structured pruning, the label "all" refers to the original, unpruned parameters, while "everyother" denotes a simple pruning where every other LoRA layer is dropped. Other labels indicate the portion of parameters retained after pruning. We also compare with the "Early CopRA", which refers to the checkpoint taken in the first quarter of the training epochs. This comparison validates our hypothesis that the earlier training stages exhibit LMC. For unstructured pruning, the sparsity refers to the percentage of parameters removed. For instance, with a pre-defined sparsity ratio of 0.1, we set 10% of the weights to zero. Our findings demonstrate that CopRA performs effectively under both pruning strategies, benefiting from its early-stage training. Furthermore, CopRA can be successfully combined with other quantization and pruning schemes. This allows each layer's parameters trained by CopRA to individually express their capabilities more effectively compared to LoRA.

### 3.3  Analysis

A reasonable assumption is that if the weights trained from different seeds are similar, the performance of merging would improve. However, the LoRA parameters produced by CopRA are not necessarily more similar. As illustrated in Fig 5(a), we visualize both LoRA and CopRA parameters using t-SNE [21] across different seeds. In addition, CopRA consistently outperforms LoRA across various learning rates, as shown in Fig. 5(b), where LoRA training fails when the learning rate exceeds 1e-3. Finally, to demonstrate the effective of CopRA in different domains, we conduct the merging tasks on MTL15 [17], which consists of 15 text classification tasks, and present the results in Fig. 5(c).

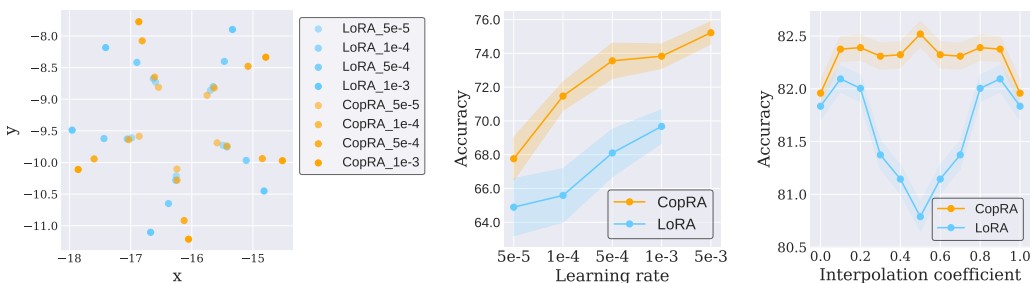

Figure 5: (left) **T-SNE visualization with different seeds and learning rates.** (middle) **Merging accuracy across different learning rates.** (right) **Merging results on the *MTL15* dataset.**

We further investigate the impact of reducing the number of iterations on CopRA. Firstly, because CopRA optimizes only a subset of parameters at each step, it trains faster with the same iterations. In CLIP-LoRA [24], the default number of iterations is set to n_iters = 500. As illustrated in Fig. 6(a), we report the average accuracy of models trained with different iterations and learning rates. Notably, CopRA exhibits underfitting with a learning rate of 5e-5 and 100 iterations, while LoRA experiences instability with a learning rate of 5e-3 and 400 iterations; therefore, these cases are not included in the results. In Fig. 6(b), under the settings described in Sec. 3.1, we present the accuracy of the merged models with an interpolation ratio of 0.5, which significantly outperforms LoRA even when using very few iterations. Additionally, Fig. 6(c) presents the merging results for the optimal configurations of both LoRA (lr=1e-3, n_iters = 400) and CopRA (lr=5e-3, n_iters = 400) across the training steps.

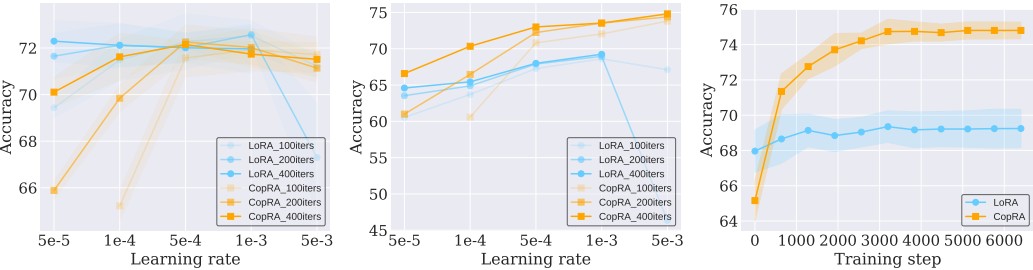

Figure 6: **Accuracy** (left) **and merging accuracy** (middle) **with different iterations and learning rates.** (right) **Accuracy over steps for the merged model,** using best results from LoRA and CopRA.

## 4  Conclusion

In this work, we introduce CopRA, a novel LoRA training strategy that enhances the standard LoRA by incorporating *linear mode connectivity* through progressive training with an increasing subset of optimized modules. This strategy aligns with optimizing the Shapley value of each layer, which implies that each LoRA enhances its marginal contribution. Notably, similar to dropout, our method is simple and effective, without incurring additional training time. Our experiments demonstrate that CopRA not only enables efficient model merging for applications in federated and multi-task learning but also achieves superior performance in pruning. In the future, we will further theoretically validate CopRA's superiority and extend its application to instruction tuning tasks on large language models.

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
