# OpenReview forum: "CopRA: A Progressive LoRA Training Strategy"
_NeurIPS.cc/2024/Workshop/UniReps — UniReps_

### Official Review · Reviewer_fLbc · 2024-09-27
**Evaluating CopRA: A Novel Progressive Training Strategy for LoRA with Cooperative Layer Optimization**

**Rating:** 8
**Confidence:** 1

**Review:**

Quality:
The overall quality of the paper is strong. The authors provide a clear problem definition—improving Low-Rank Adaptation (LoRA) to address limitations in merging, pruning, and generalization—and propose a well-motivated solution through CopRA. The methodology is presented with sufficient detail, supported by theoretical grounding and cooperative game theory, making the approach sound and justifiable. The experimental results are thorough and demonstrate the advantages of CopRA across multiple tasks and datasets, adding credibility to the claims made. However, some aspects, particularly regarding computational complexity and potential trade-offs of the proposed method, could have been discussed more explicitly.

Clarity:
The clarity of the work is generally good, but there is room for improvement in certain areas. Key concepts like the Shapley value, layer-wise linear mode connectivity, and their integration into the CopRA framework are explained clearly. Visual aids such as graphs, tables, and figures further support understanding. However, the introduction of CopRA’s training strategy and its connection to the cooperative game theory could be made more accessible to a broader audience by simplifying some explanations. Additionally, while the authors touch on practical aspects like merging and pruning, a more detailed analysis of how CopRA compares in terms of time and computational resources to baseline methods could enhance the clarity of its practical impact.

Originality:
The paper introduces original contributions, particularly in the novel combination of progressive training and cooperative game theory for LoRA fine-tuning. The progressive layer-dropping approach, inspired by dropout techniques but specifically adapted to LoRA, represents a novel application that hasn't been widely explored in parameter-efficient training methods. The use of Shapley value optimization to evaluate each layer’s contribution in the context of model merging and federated learning is an innovative twist on cooperative game theory applied to neural networks. While the underlying concepts of LoRA and dropout are not new, the specific way they are integrated and enhanced in CopRA adds a unique dimension to the current literature.

Significance:
The significance of this work lies in its potential impact on several key areas of AI, particularly in federated learning and multi-task learning (MTL), which are crucial for decentralized and resource-efficient AI models. By improving LoRA’s generalization capabilities and enabling more effective model merging, CopRA addresses practical challenges in these fields. Additionally, the demonstrated effectiveness in pruning enhances its relevance for model compression, which is increasingly important in deploying large models on edge devices or in environments with limited resources. If adopted and further refined, CopRA could have broad implications for how large models are fine-tuned and maintained in federated systems.

Pros:
Novel Training Strategy: The progressive layer-dropping approach is innovative and offers a fresh way to handle parameter-efficient training.
Cooperative Game Theory Application: The integration of Shapley value to optimize each layer’s contribution introduces a novel perspective on fine-tuning models.
Enhanced Model Merging: CopRA improves linear mode connectivity, enabling better performance in federated learning and multi-task learning scenarios.
Efficiency in Pruning: The method offers effective pruning strategies that reduce the number of parameters while maintaining accuracy, a valuable asset for model compression.
Comprehensive Experimental Results: The experiments are extensive, covering multiple datasets and showing the superiority of CopRA in merging and pruning compared to standard LoRA.
Cons:
Limited Discussion on Computational Overhead: The potential increase in computational complexity due to progressive training and Shapley value calculations is not fully addressed.
Clarity in Explanations: While the paper is generally clear, some of the more technical explanations, particularly around cooperative game theory, could be simplified for broader understanding.
Scalability Concerns: The paper does not discuss how well CopRA scales to extremely large models or datasets, which could be important for practical adoption.
Absence of Real-World Benchmarking: While the paper offers solid theoretical and experimental insights, real-world applications or large-scale benchmarking would enhance the practical relevance of the results.
Comparison to Other Efficient Training Methods: The paper focuses heavily on comparing CopRA to standard LoRA but does not explore how it fares against other parameter-efficient methods like adapters or distillation techniques.
Conclusion:
Overall, this work presents a compelling and innovative approach to enhancing LoRA with the progressive CopRA training strategy. It balances theoretical contributions with practical applications, making it a significant addition to the field of parameter-efficient training. The strengths of the work lie in its originality, the novel use of cooperative game theory, and its practical implications for federated learning, multi-task learning, and model pruning. However, further discussion on the method's computational cost and scaling potential would help solidify its broader impact.

---

### Official Review · Reviewer_53nv · 2024-10-04
**Decision on the Rejection of Manuscript: 'CopRA: A Progressive LoRA Training Strategy**

**Rating:** 3
**Confidence:** 4

**Review:**

Dear Author(s),

While your work presents an interesting method to LoRA training, there are several concerns that prevent us from accepting the paper in its current form.

Strengths:
The paper introduces a novel progressive training strategy for LoRA, which is a timely and relevant topic in the field of efficient fine-tuning for large language models.
The proposed CopRA method shows promising results in model merging tasks, demonstrating potential applications in federated learning and multi-task learning scenarios.
The approach of viewing LoRA layers as players in a cooperative game and optimizing Shapley values is innovative and provides an interesting perspective on the problem.

Weaknesses:
Lack of theoretical justification: While the paper presents an intuitive explanation for the proposed method, it lacks a rigorous theoretical foundation to support the claims about linear mode connectivity and Shapley value optimization.
Limited experimental validation: The experiments are primarily focused on image classification tasks using CLIP. More diverse tasks and model architectures would strengthen the paper's claims about the generalizability of the method.
Insufficient comparison with state-of-the-art: The paper does not adequately compare CopRA with other recent advancements in LoRA training and merging techniques and CopRA is simple idea, making it difficult to assess its relative contribution to the field.
Clarity and presentation issues: Some parts of the paper, particularly in the methodology section, are unclear and could benefit from improved explanations and illustrations.
Reproducibility concerns: The paper lacks sufficient details about hyperparameters and implementation specifics, which may hinder reproducibility of the results.
We encourage the authors to address these concerns and consider submitting a revised version to a future conference or journal. Specifically, we recommend:

Strengthening the theoretical foundation of the proposed method.
Expanding the experimental evaluation to include a wider range of tasks and model architectures.
Providing a more comprehensive comparison with state-of-the-art methods in LoRA training and merging.
Improving the clarity of the methodology section and overall presentation.
Including more details to ensure reproducibility of the results.
We appreciate your contribution to the field and hope that these comments will be helpful in further developing your work.

---

### Official Review · Reviewer_tsJY · 2024-10-06
**Copra as a training strategy**

**Rating:** 6
**Confidence:** 3

**Review:**

Overall the paper is technically sound.

The mathematical equations seem to be correct and understandable.
However, in my view, some things need to be explained more properly
Figure 4: needs to be explained properly, can you explain sparsity more correctly?

The connection to federated learning
Also if there have already been methods to surpass LoRA. How is your method super different?

---

> ### Author Response · Authors · 2024-10-14
> **Some clarifications**
>
> We appreciate the reviewer's insightful comments and would like to address them as follows:
>
> #### [Q.1] Can you explain sparsity in Figure 4 more correctly?
>
> > The sparsity in Figure 4 refers to the percentage of parameters removed during unstructured pruning. For instance, with a pre-defined sparsity ratio of 0.1, we set 10% of the weights to zero. We apologize for not providing clearer details in the original submission and will update the camera-ready version with a more thorough explanation.
>
> #### [Q.2] The connection to federated learning Also if there have already been methods to surpass LoRA. How is your method super different?
>
> > Our method primarily enhances LoRA by introducing a novel training strategy, offering advantages in aspects like merging. In contrast, existing federated learning approaches [14, 23, 25] focus on innovations in the merging process for existing LoRAs rather than improving the training process.
> >
> > This distinction means CopRA can potentially be combined with those methods. We appreciate your suggestion and will ensure to expand on this in our future revisions.

---

### Official Review · Reviewer_12LE · 2024-10-07
**The proposed training strategy enhances LoRA by promoting linear mode connectivity, improving model fusion and generalization.**

**Rating:** 8
**Confidence:** 3

**Review:**

### **Strengths:**

1. **Systematic Activation for Efficient Exploration**:
   The paper introduces a gradual increase in LoRA module activation probability, which allows for an efficient exploration of diverse optima during the initial training phase, ultimately converging to a well-refined solution. This approach enhances model generalization and ensures that the final model retains the potential for linear interpolation between different solutions, as supported by the empirical results. The systematic activation presents a novel and effective mechanism for improved exploration of the optimization landscape.

2. **Use of Cooperative Game Theory**:
   By framing each LoRA layer as a player in a cooperative game, the paper leverages the Shapley value to assess the contributions of individual layers in a fair and structured manner. This approach provides insightful metrics on the significance of each layer's impact on overall model performance, which is instrumental in guiding strategic optimization. Furthermore, this framework also strengthens pruning capabilities by effectively reducing parameter redundancy, ultimately enhancing the model's efficiency without sacrificing performance.

### **Weaknesses:**

1. **Increased Training Complexity and Duration**:
   The proposed strategy, involving a progressive adjustment of activation probabilities across LoRA modules, introduces additional complexity into the training pipeline. This multi-stage optimization process is likely to extend the training duration compared to conventional LoRA training. A more detailed analysis of the resulting training time overhead, including quantitative comparisons, would provide valuable insights and allow for a more complete assessment of the trade-offs involved.

2. **Transition from Stochastic to Deterministic Activation**:
   In the final quarter of the training cycle, when all LoRA modules are activated, the model shifts from a stochastic regime to a deterministic one. While this aligns the final training phase with standard LoRA objectives, a potential concern is whether such a sudden shift might introduce a sub-optimal transition, potentially undermining the model's performance? Given my limited knowledge here, this is a point that  I would be particularly interested in understanding whether this transition affects the stability or quality of the resulting solutions. A analysis is would be a good idea.

### **Overall Assessment**:

The paper proposes a solid and innovative framework that addresses key challenges in model merging, fusion, and pruning—areas where LoRA has notable limitations. The novel techniques presented are well-founded and demonstrate potential for significant impact, particularly with respect to linear mode connectivity and parameter efficiency. However, there are areas where more extensive experimentation and theoretical validation could strengthen the findings, specifically regarding the training duration implications and the stochastic-to-deterministic transition. Nonetheless, given the apparent preliminary nature of this work and the scope of the extended abstract, I find the contributions promising and well worth further exploration.

---

> ### Author Response · Authors · 2024-10-30
> **Some clarifications**
>
> We appreciate the insightful feedback provided by the reviewer and would like to address the concerns as follows:
>
> #### [W.1] Increased Training Complexity and Duration
>
> > In fact, under the same training steps, our strategy (CopRA) accelerates the training speed of LoRA by skipping the training of certain layers, similar to the method of subnetwork training mentioned in [r1]. Additionally, the benchmark defaults to 500 iterations, we have conducted supplementary experiments at 100, 200, and 400 iterations. The results, depicted in Figure 6 (middle), show that CopRA maintains robust performance even at 100 iterations with higher learning rates, clearly demonstrating that our method does not increase the training duration.
>
> #### [W.2] Transition from Stochastic to Deterministic Activation
>
> > During the final training phase, our objective is to closely align with direct training strategies to just enhance individual model accuracy, as depicted in Figure 6 (left). This approach ensures the utilization of each layer's LoRA during final inference, contrasting with our initial phase strategy where some layers were randomly skipped. Additionally, as illustrated in Figure 6 (right), there is no reduction in performance on model merging tasks in the last quarter of the epoch.
>
> #### Reference
>
> [r1] Li, Hanqi, et al. "Evolving Subnetwork Training for Large Language Models." ICML 2024.

---

### Decision · Program_Chairs · 2024-10-10

**Decision:**

Accept

**Comment:**

In light of the positive reviewers' feedback and relevancy of the submission, we are pleased to accept this paper for presentation at UniReps 2024. We kindly ask the authors to incorporate the reviewers' suggestions and feedback in the final camera-ready version of the manuscript.